# Rapid Detection of SARS-CoV-2 Based on the LAMP Assay Associated with the CRISPRCas12a System

**DOI:** 10.3390/diagnostics13132233

**Published:** 2023-06-30

**Authors:** Karoline Almeida Felix de Sousa, Carolina Kymie Vasques Nonaka, Ricardo Khouri, Clarissa Araújo Gurgel Rocha, Carlos Gustavo Regis-Silva, Bruno Solano de Freitas Souza

**Affiliations:** 1Gonçalo Moniz Institute, Oswaldo Cruz Foundation (FIOCRUZ), Salvador 40296-710, Brazil; karool.felix@gmail.com (K.A.F.d.S.); ricardo_khouri@hotmail.com (R.K.); clarissa.gurgel@fiocruz.br (C.A.G.R.); 2Center for Biotechnology and Cell Therapy, D’Or Institute for Research and Education (IDOR), Salvador 41253-190, Brazil; carolina.nonaka@hsr.com.br; 3São Rafael Hospital, Salvador 41253-190, Brazil; 4Faculty of Medicine, Federal University of Bahia, Salvador 40210-630, Brazil; 5Faculty of Dentistry, Federal University of Bahia, Salvador 40210-630, Brazil

**Keywords:** COVID-19, point-of-care, CRISPR, infection control, diagnostic test

## Abstract

Background: The global public health system has been severely tested by the COVID-19 pandemic. Mass testing was essential in controlling the transmission of the SARS-CoV-2; however, its implementation has encountered challenges, particularly in low-income countries. The urgent need for rapid and accurate tests for SARS-CoV-2 has proven to be extremely important. Point-of-care tests using the CRISPR system for COVID-19 have shown promise, with a reported high sensitivity and rapid detection. The performance of a CRISPR-based SARS-CoV-2 testing system was reported in this study. Methods: A total of 29 nasopharyngeal samples were evaluated, including 23 samples from individuals suspected of COVID-19, and six samples positive for H3N2 or respiratory syncytial virus. Two reference samples with known concentrations of SARS-CoV-2 RNA (3000 RNA copies/mL) or viral titer determined by plaque assay (105 PFU/mL) were also evaluated. The LAMP technique was employed to amplify the ORF1ab gene and the results were analyzed using a Gemini XPS fluorescence reader. Results: The RT-LAMP-CRISPR/Cas12 assay showed 100% concordance compared to RT-PCR. The RT-PCR presented a detection limit of 0.01 PFU/mL and the CRISPR/Cas12 system showed a limit of 15.6 PFU/mL. The RT-PCR sensitivity was approximately 8 RNA copies/µL and CRISPR/Cas12 at 84 RNA copies/µL. Conclusion: The RT-LAMP-CRISPR/Cas12a assay offered a promising alternative for the detection of SARS-CoV-2 and reinforces that CRISPR-based diagnostic techniques can be an alternative for fast and accurate assays.

## 1. Introduction

In late 2019, the first cases of pneumonia caused by SARS-CoV-2 were reported in Wuhan, China [1,2]. The virus spread rapidly around the world and, in March 2020, the WHO declared a state of pandemic [3]. By January 2023, more than 670 million infections and more of six million deaths had been reported [4]. Studies reported that the probability of infection after contact with an infected person was up to 83% [1].

Accordingly, screening plans have been implemented to accurately identify SARSCoV-2 infected patients, thus reducing the risk of contagion [5]. Nucleic acid amplification is the most indicative of SARS-CoV-2 as it detects the virus in the first days of infection [6]. However, the present gold-standard detection assays using RT-PCR require the use of equipped laboratories and the transport of samples to central laboratories with a 4–6 h delay for the release of their results, reducing the prospect of rapid diagnosis [7].

The devastating effects of SARS-CoV-2 worldwide demonstrates the need for the implementation of easy diagnostic techniques to be applied in remote areas, airports, and in low-income countries with higher risks of SARS-CoV-2 transmission, and the use of rapid, portable, and low-cost equipment to control viral spread in the population [8]. Approaches have been used in order to accelerate diagnostic testing for COVID-19, reverse transcription loop-mediated isothermal amplification (RT-LAMP), detecting RNA virus in a single reaction at only a single temperature, and presenting a rapid alternative to PCR-based methods [9,10]. However, this assay has low specificity, making it difficult to use for the diagnosis of SARS-CoV-2 [11,12]. 

The U.S. Food and Drug Administration (FDA) has approved the detection of viral RNA based on CRISPR technology. This system is promising in the diagnosis of SARS-CoV2 because of its high specificity and sensitivity in detecting nucleic acid through a single Cas protein [13,14]. Therefore, we report here the development of a rapid assay using CRISPR-Cas 12-based RT-LAMP for the detection of SARS-CoV-2 RNA.

## 2. Materials and Methods

### 2.1. Ethical Statement and Samples

A convenient cohort, consisting of patients admitted to the emergency room of São Rafael Hospital (Salvador, Bahia, Brazil), was included in this study. A total of 29 clinical samples (nasopharyngeal swabs) were collected from February 2022 to March 2022. Among these samples, there were 13 positive and 16 negative cases, confirmed by RT-PCR by the laboratory within the same hospital. Samples obtained from individuals aged 18 years and above presenting symptoms consistent with COVID-19 were collected and subsequently confirmed by RT-PCR analysis. Negative samples were also confirmed by RT-PCR. Prior to participation, written consent was obtained from all enrolled patients, and this study received approval from the Institutional Review Board (Number CAAE: 38580920.6.3001.0040). Samples were submitted to RNA extraction using the viral kit RNA 93 MagMAX™ Viral/Pathogen II (MVP II) Nucleic Acid Isolation Kit (ThermoFisher Scientific, Waltham, MA, USA), according to the manufacturer’s recommendations. Confirmation of positive and negative samples was performed with the Allplex™ SARS-CoV-2 kit (Seegene^®^, Seoul, Republic of Korea) according to the manufacturer’s recommendations and the thermal cycler ABI 7500 FAST (Applied Biosystem, Waltham, MA, USA). To establish the limit of detection (LoD) of Cas12a, a 1:10 serial dilution was used in nuclease-free water of titration SARS-CoV-2 (1.5 × 10^5^ PFU), provided by the virology laboratory of the Federal University of Minas Gerais and so was a reference sample with 3000 RNA copies/mL (AccuPlex SARS-CoV-2 reference, Sera Care, Milford, MA, USA). 

### 2.2. Nucleic Acid Preparation

SARS-CoV-2 genome sequences were obtained from the GISAID website (www.gisaid.org) (accessed on 5 October 2020) and lined up to obtain a consensus sequence. Possible compatible sites for the construction of guide RNA were located throughout the sequence; subsequently, oligonucleotides with regions of the N gene and the ORF1ab gene were designed using PrimerExplorer v.5 (https://primerexplorer.jp/e/) (accessed on 5 October 2020) according to instructions in the guide published by the site’s developers. Selected regions were then used in “nucleotide versus nucleotide” searches on the website https://blast.ncbi.nlm.nih.gov/Blast.cgi (accessed on 5 October 2020) to check for homology with sequences of coronaviruses that infect humans. For control, the RNAse P gene was used as a target, as previously described [11]. All the primers and RNAgs used in this study are listed in Appendix A.

### 2.3. Cas12a Detection Reactions

Fragments of ORF1ab, N gene of SARS-CoV-2 and RNAse P gene were synthesized and amplified using the RT-LAMP assay, following the instructions of the manufacturer, New England Biolabs Inc. (https://www.neb.com/protocols/2014/10/09/typical-rt-lamp-protocol) (accessed on 2 May 2022). The reactions were performed individually for the detection of each gene, using MgSO_4_ and dNTP mix at final concentrations of 8mM and 1.4mM, respectively. A total of 7 µL of RNA was used, heating the final reaction at 65 °C for 30 min. 

The optimized trans-cleavage assays were performed as described previously [11,12]. We used approximately 30 nM of LbCas12a (NEB) and incubated for 10 min with 40 nM gRNA in addition to NEBuffer 2.1 at 37 °C. Subsequently, 2 µL of amplicon was added and after the RNA-protein complex’s formation, 100 nM of the fluorescent reporter 6-FAMBHQ was added to a black 384-well plate and readings were performed in the Gemini XPS fluorescence microplate.

### 2.4. Statistical Analysis

Prism Software v.9.1.1 (GraphPad, La Jolla, CA, USA) was used to analyze all the data. Cohen’s Kappa coefficient was used. Data regarding Ct values and fluorescence levels from RT-PCR and RT-LAMP/CRISPR-CAS12a, respectively, were analyzed by Pearson correlation (r). The summary of the methodology is illustrated in the Figure 1.

## 3. Results

Initially, we conducted an assessment of different primer sets and SARS-CoV-2 targets in RT-LAMP reactions to determine their performance. Following the evaluation and standardization of the oligonucleotides, we observed satisfactory amplification of fragments from the ORF1ab gene (G2) of SARS-CoV-2 through LAMP (refer to Appendix A for details). We subsequently employed the RT-LAMP-CRISPR/Cas12a assay with fluorescence-based readings to analyze the RNA extracted from the 29 respiratory swab samples. Of those samples, 16 tested negative for SARS-CoV-2. Among these 16 negative samples, six were positive for H3N2 or respiratory syncytial virus. Among the patient swabs tested, 13 were found to be positive for SARS-CoV-2, and no cross-reactions with other respiratory viruses were observed. Remarkably, the results obtained from the RT-LAMP-CRISPR/Cas12a assay exhibited 100% concordance with those obtained from the gold-standard RT-PCR protocol with a Kappa value of 1. (see Figure 2). To ensure the reliability of the RT-LAMP-CRISPR/Cas12a assay, we incorporated the human RP gene as an internal control.

Furthermore, we compared the analytical limit of detection (LoD) between the LAMP-CRISPR/Cas12a assay and RT-PCR. Serial dilutions of titrated SARS-CoV-2 RNA (1.5 × 10^5^ PFU/mL) were performed. The commercially available RT-PCR kit demonstrated a detection limit of 0.15 PFU/mL (dilution 10^−6^), while the RT-LAMP-CRISPR/Cas12a assay was able to detect a limit of 15.6 PFU/mL (dilution 10^−4^) (see Figure 3).

Subsequently, we compared the analytical limit of detection (LoD) of the LAMP- CRISPR/Cas12a assay and RT-PCR. Serial dilution of titration SARS-CoV-2 RNA (1.5 × 10^5^ PFU/mL) was performed, the RT-PCR methodology using the commercial kit presented a detection limit of 0.15 PFU/mL (dilution 10^−6^) and the RT-LAMP-CRISPR/Cas12a assay was able to detect a limit of 15.6 PFU/mL (dilution 10^−4^) (Figure 3).

To determine the sensitivity of RT-LAMP-CRISPR/Cas12a, we used a reference sample with 3000 RNA copies/mL (AccuPlex SARS-CoV-2 reference, Sera Care). The RT-PCR method demonstrated a sensitivity of approximately 8 RNA copies/µL (point 10^−1^) and the LAMP-CRISPR-Cas12 demonstrated 84 RNA copies/µL (point 10^0^), as illustrated in Figure 4.

## 4. Discussion

During the pandemic, a critical mechanism for delaying SARS-CoV-2 dissemination was to implement testing for the detection and isolation of cases as early as possible [15]. The RT-LAMP (reverse transcription loop-mediated isothermal amplification) assay reported here is a fast and simple assay, avoiding the need for thermal cycling and the use of complex laboratory infrastructure. It could easily be used at critical points of COVID-19 transmission to simplify the diagnostic process. 

Some studies have highlighted the use of the LAMP technique in detecting coronavirus infections in patient samples [16,17]. RT-LAMP-based assays demonstrated an almost 90% sensitivity and high consistency compared to RT-PCR-based diagnostic methods [18]. The time required for this assay was about one hour, considerably less than for RT-PCR. The assay in this RT-LAMP-CRISPR/Cas12a study was performed in only 30 min. According to Hong Thai and collaborators, the RT-LAMP assay demonstrated higher sensitivity than conventional RT-PCR, with a detection limit of 0.01 PFU in clinical samples [19].

One limitation of diagnostic assays is their inability to consistently detect low levels of the virus in a sample, which can result in false-negative results, particularly during the early stages of infection or when the viral load is low. In comparison to other detection methods, the gold standard RT-PCR exhibits higher sensitivity in this regard. Another challenge is that the SARS-CoV-2 has shown the ability to mutate and give rise to new variants. CRISPR-based diagnostic assays often rely on the detection of specific genomic regions or sequences. If these regions undergo mutations or variations in the viral genome, it may affect the accuracy of the test. Continuous monitoring and updating of the CRISPR system to detect emerging variants would be necessary. Future advancements may help overcome these challenges and improve the feasibility and effectiveness of CRISPR-based diagnostics for SARS-CoV-2 and other infectious diseases.

CRISPR (clustered regularly interspaced short palindromic repeats) has become important in molecular biology experiments in recent years, reformulating the diagnoses of the present moment. Numerous CRISPR-based techniques have been developed as potential alternatives for diagnosing SARS-CoV-2, including SHERLOCK, and other lateral flow-based diagnostics [20].

This study demonstrated an alternative for the sensitive detection of SARS-CoV-2 by a rapid LAMP assay together with the CRISPR-Cas12a system. The methodology and results of this study demonstrated the ease of use, effectiveness, efficiency, accuracy, and satisfaction in the use of the technology using RT-LAMP-associated Cas12a through fluorescence detection in the diagnosis of SARS-CoV-2. This methodology could also be adapted to a lateral flow strip-based detection system. Studies reported that the use of the lateral flow strip showed a sensitivity of more than 90% compared with the gold standard RT-PCR test [21]. Finally, although the RT- LAMP/CRISPR-based method described herein displays advantages in terms of protocol duration and no requirements of molecular biology infrastructure, the test sensitivity is significantly inferior to RT-PCR, requiring further development. The LoD is situated within the range of commercial RT-PCR and may be sufficient for the detection of SARS-CoV-2 in most clinical samples [22]. Moving forward, the detection method described in this study has the potential for adaptation to target various respiratory pathogens. By optimizing the development of rapid and targeted assays, this approach can be modified to detect and identify other respiratory pathogens effectively. The flexibility of the described detection technique allows for its future application in diagnosing a broader range of respiratory infections, offering a promising avenue for further research and advancements in the field.

## 5. Conclusions

The RT-LAMP-CRISPR/Cas12a assay presents a promising and rapid alternative for the detection of SARS-CoV-2. Its high concordance with RT-PCR, along with its acceptable sensitivity and detection limit, suggests its potential usefulness in diagnosing COVID-19 cases. This CRISPR-based system holds promise for enhancing global testing efforts, particularly in resource-limited settings.

## Figures and Tables

**Figure 1 diagnostics-13-02233-f001:**
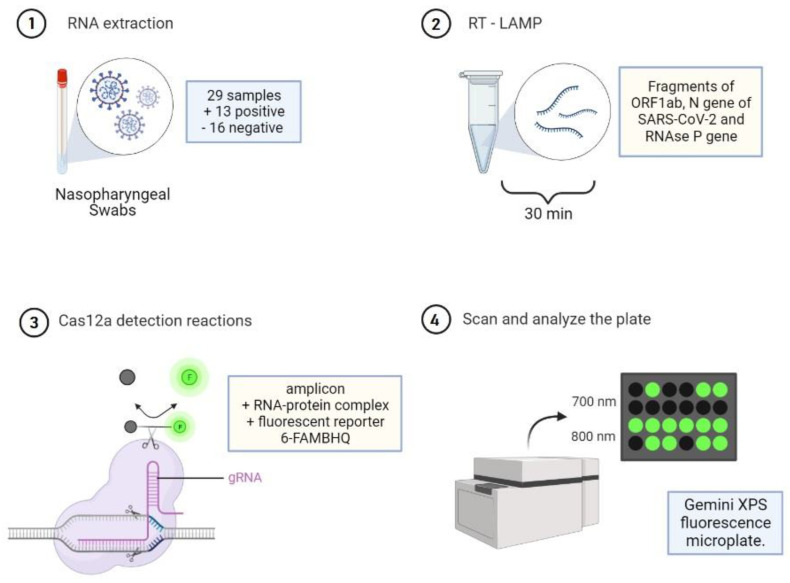
Schematic representation of the study. Green = detected and Black = not detected.

**Figure 2 diagnostics-13-02233-f002:**
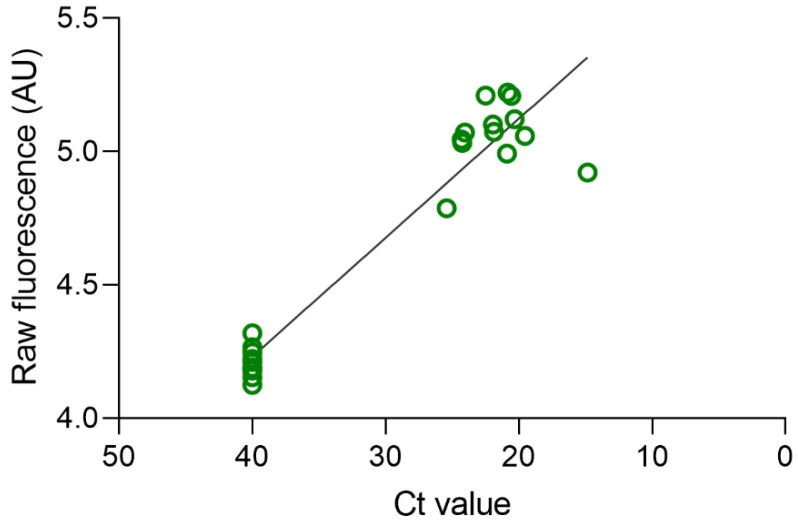
Correlation of the Ct values obtained by RT-PCR and fluorescence level obtained from the LAMP/CRISPR assay in 29 samples. *X*-axis Ct values (RT-PCR) and *Y*-axis raw fluorescence values (RT-LAMP-CRISPR/Cas12a). R^2^ = 0.9176.

**Figure 3 diagnostics-13-02233-f003:**
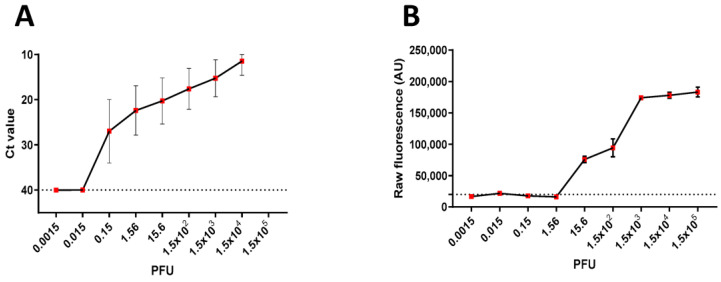
Comparison of the limit of quantification of titration SARS-CoV-2 virus (point 100 = 1.5 × 10^5^ PFU) between methodologies. (**A**) RT-PCR using a commercial kit (detected < Ct 40, triplicate by spot). (**B**) RT-LAMP-CRISPR/Cas12a protocol (detected > 20,000 AU, quintuplicate by spot).

**Figure 4 diagnostics-13-02233-f004:**
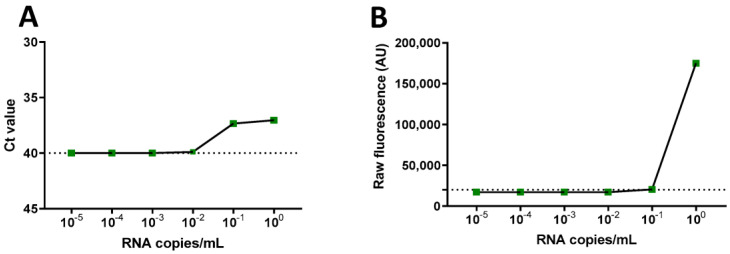
Determination of the sensitivity of the CRISPR/Cas12a assays by RT-PCR (**A**) and RT-LAMP (**B**), in values of viral RNA copies/mL.

## Data Availability

Not applicable.

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
