# Peer review of "Rapid Detection of SARS-CoV-2 Based on the LAMP Assay Associated with the CRISPRCas12a System"

_diagnostics, 2023, doi:10.3390/diagnostics13132233_

Round 1
Reviewer 1 Report
The authors developed a detection system for SARS-CoV2 identification based on RT-LAMP-Cas12. The research has a clear design, methods are well described and can be reproduced. Results are reported explicitly. The main drawback of the paper is absence of comparison with the other LAMP-Cas12 methods of SARS-CoV2 detection (e.g. 10.3390/diagnostics11091646, 10.1016/j.snb.2021.130411, 10.1016/j.virusres.2020.198129, 10.3389/fmed.2021.627679, etc.). Authors should add a table with LoD, time of analysis, other characteristics of these methods, and the developed one.
Abstract: Please, highlight the novelty of your research.
Line 54. Please provide a reference.
Line 72: I recommend mentioning molar concentration of the RNA as well
Fig. 1. Enumeration of “Scan and analyze the plate” might be incorrect - #3 instead of #4
Line 122, 123: I would write “dilution 106”/”dilution 104” instead of dilution 10 -6”/”dilution 10-4”
Line 125-127 “…100% concordance of the RT-LAMP-CRISPR/Cas12a assay results compared to the gold standard RT-PCR protocol (Figure 2). The RP gene was used as an internal control for the RT-LAMP-CRISPR/Cas12a assay.” is a repeat of Line 115-117
Lines 128-134 are the duplication of Lines 119-123.
Table S2. You mentioned the reporter for lateral flow assay. However, no description of one in the manuscript. If the LFA was not implemented, please remove the reporter from Table S2.
I have not found typos or incorrect terms. The grammar of the manuscripts seems correct. Perhaps, minor proofreading can improve the paper.
Author Response
We expressed our gratitude to the reviewer for their valuable contributions and implemented the recommended changes as suggested. Please refer to the attached document for the requested answers.
The authors developed a detection system for SARS-CoV2 identification based on RT-LAMP-Cas12. The research has a clear design, methods are well described and can be reproduced. Results are reported explicitly. The main drawback of the paper is absence of comparison with the other LAMP-Cas12 methods of SARS-CoV2 detection (e.g. 10.3390/diagnostics11091646, 10.1016/j.snb.2021.130411,
10.1016/j.virusres.2020.198129, 10.3389/fmed.2021.627679, etc.). Authors should add a table with LoD, time of analysis, other characteristics of these methods, and the developed one.
Abstract: Please, highlight the novelty of your research.
Answer: We appreciate the suggestion and have incorporated it into the final conclusion, stating that the CRISPR tool can serve as a rapid and precise alternative for diagnostic tests. Although our data focuses on its application for SARS-CoV-2, it is essential to highlight that this technique has the potential to be utilized for other targets as well.
Line 54. Please provide a reference
Answer: A reference number 7 was inserted
Line 72: I recommend mentioning molar concentration of the RNA as well
Answer: We clarify that the RNA concentration was not measured. The automated
extraction technique was validated to be used in molecular diagnosis by RT-PCR or for genomic sequencing, following the manufacturer's recommendations.
Fig. 1. Enumeration of “Scan and analyze the plate” might be incorrect - #3 instead of #4
Answer: Thank you very much for alerting us. We have already corrected the figure.
Line 122, 123: I would write “dilution 106”/”dilution 104” instead of dilution 10 -
6”/”dilution 10-4”
Line 125-127 “…100% concordance of the RT-LAMP-CRISPR/Cas12a assay results
compared to the gold standard RT-PCR protocol (Figure 2). The RP gene was used as an internal control for the RT-LAMP-CRISPR/Cas12a assay.” is a repeat of Line 115-117
Lines 128-134 are the duplication of Lines 119-123.
Table S2. You mentioned the reporter for lateral flow assay. However, no description of one in the manuscript. If the LFA was not implemented, please remove the reporter from Table S2.
Answer: Thank you very much for bringing this to our attention. We have already
corrected the writing.
-----------------------------------------------------------------------
Reviewer 2 Report
Title: Indicate the type and place of study
Abstract: needs to be concise and specific.
Keywords: should not be the same as the words used in the title.
Include the following suggestions to strengthen the introduction and discussion segment of the manuscript:
1. Introduction on COVID-19 with emphasis on its origin and transmission (refer and cite: doi: 10.1136/postgradmedj-2020-138234).
2. With the advent of vaccination that has curtailed the global Covid-19 cases, manuscripts should mention about: t he role of vaccination in one or two statements (refer and cite: doi: 10.3390/vaccines9101064),
General points:
1. Inclusion and exclusion criteria to be included.
2. Where there any missing cases? If yes how was it considered during statistical analysis.
3. When the study factors were divided into dependent and independent variable, how was the bias managed during the statistical evaluation?
4. Ethical certificate number should be mentioned.
5. Strengths and limitations to be included
6. Include a Forrest plot or more diagrammatic representations of the statistical indices.
Minor grammatical errors need to be corrected
Author Response
We expressed our gratitude to the reviewer for their valuable contributions and implemented the recommended changes as suggested. Please refer to the attached document for the requested answers.
Title: Indicate the type and place of study
Abstract: needs to be concise and specific.
Keywords: should not be the same as the words used in the title.
Answer: We sincerely appreciate the suggestion, and we have duly incorporated the recommended adjustments into the abstract and keywords.
Include the following suggestions to strengthen the introduction and
discussion segment of the manuscript:
1. Introduction on COVID-19 with emphasis on its origin and transmission (refer and cite: doi: 10.1136/postgradmedj-2020-138234).
2. With the advent of vaccination that has curtailed the global Covid-19 cases,
manuscripts should mention about: t he role of vaccination in one or two
statements (refer and cite: doi: 10.3390/vaccines9101064),
Answer: We appreciate the suggestion and inform you that one of the suggested
references was cited in the manuscript.
General points:
1. Inclusion and exclusion criteria to be included.
Answer: We inform you that the paragraph of item 2.1 has been rewritten to clarify the inclusion and exclusion criteria.
2. Where there any missing cases? If yes how was it considered during statistical analysis.
Answer: We had no missing cases. 29 samples were included and all were represented. To provide clarification on this matter, the study included a total of 13 positive samples and 13 negative samples. However, it is important to note that within the negative sample group, we included samples that tested negative for SARS-CoV-2 but positive for other viruses. This was done in order to assess the specificity of the assay. As a result, there was a greater number of samples in the negative group (16 samples) as compared to the positive group (13 samples).
3. When the study factors were divided into dependent and independent variable, how was the bias managed during the statistical evaluation?
Answer: We appreciate the reviewer's comments. To clarify, the concordance test was used in this study. We include measurement information via Cohen's kappa coefficient (line 136).
4. Ethical certificate number should be mentioned.
Answer: The project registration number was presented as mentioned in the methodology (CAAE: 38580920.6.3001.0040) CAAE: Certificate of Presentation of Ethical Appreciation. To clarify this paragraph was rewritten.
5. Strengths and limitations to be included
Answer: A paragraph was insert between lines 177 to 187
6. Include a Forrest plot or more diagrammatic representations of the statistical indices.
Answer: The median with SD was represented in figure 3. This figure illustrates the
sensitivity found in each technique using the same standard curve. In this study, a novel test was evaluated in comparison to the gold standard method using RT-PCR. Our primary objective did not entail investigating or contrasting our findings with other tests utilizing the same technology. Therefore, the concordance test was used as already mentioned in question 3.